# Thunder: a Fast Coordinate Selection Solver for Sparse Learning

**Shaogang Ren, Weijie Zhao, Ping Li**
Cognitive Computing Lab
Baidu Research
10900 NE 8th St. Bellevue, WA 98004, USA
{shaogangren, weijiezhao, liping11}@baidu.com

## Abstract

$\ell_1$ regularization has been broadly employed to pursue model sparsity. Despite the non-smoothness, researchers have developed efficient algorithms by leveraging the sparsity and convexity of the problem. In this paper, we propose a novel active incremental approach to further improve the efficiency of the solvers. We show that our method performs well even when the existing methods fail due to the low sparseness or high solution accuracy request. Theoretical analysis and experimental results on synthetic and real-world data sets validate the advantages of the method.

## 1 Introduction

$\ell_1$ regularization has been used broadly in problems such as sparse learning, compressed sensing, and so on [31, 5, 7, 6, 4, 37]. By leveraging the sparsity, people have developed many methods to scale up the solutions of $\ell_1$ regularized problems. Assume $n$ data samples with $p$ features form an $n \times p$ design matrix $X$, with an $n \times 1$ label vector, $\mathbf{y}$. Let the general loss function $f$ in (1) be a convex function with $L$-Lipschitz gradient. The primal of $\ell_1$ regularized problems is given as

$$P: \ \min_{\beta} \sum_{j=1}^{n} f(x_{j\bullet}\beta, y_j) + \lambda||\beta||_1. \tag{1}$$

Here $\lambda > 0$ is the regularization hyper-parameter. $x_{j\bullet}$ is the $j$-th row (vector) of $X$. (Later we will denote $x_i$ the $i$-th column (vector) of $X$.) $\beta \in R^{p \times 1}$ is model parameter vector. We first review the existing methods for $\ell_1$ convex problems, and then briefly state our contributions.

### 1.1 Sequential and Dynamic Screening

The basic idea of screening method is to use an approximate solution (usually the dual variable) to estimate the activity (the likelihood of the corresponding entry in model fitting parameter is non-zero) of each feature. There are two main categories of screening methods for sparse models: sequential and dynamic screening methods. Sequential screening requires to solve a sequence of sparse learning problems corresponding to a sequence of descending model penalty parameters to tighten the range estimates of the dual variable to achieve the high screening power. The strong rule [32] derives the sequential screening rule based on the assumption that the absolute values of the inner products between features and the residue are non-expansive with respect to the parameter values. The sequential screening methods proposed by [13, 24, 35, 33, 28, 34] do not take the unsafe assumptions that the strong rule uses, but try to develop safe feature screening rules based on the structure of the problem. Such a sequential procedure is suitable and efficient when solving a sequence of sparse learning problems with different regularization parameters.

Sequential screening methods are not absolutely safe since most of these methods do not consider the safe duality gap, which is pointed out by the authors of dynamic screening methods [21]. Instead of using the solution information from a heavier parameter, dynamic screening methods [2, 9, 21, 22] rely on many iterations of sub-gradient computation regarding the whole feature set to gain a small duality

gap. The computation cost of these operations dilutes the screening benefits as the iterations have to be repeated many times to arrive at a sufficient small duality gap to achieve desired screening power.

## 1.2 Homotopy Method

Homotopy methods have been applied for sparse models to compute the solution path when $\lambda$ varies [25, 8, 18, 12, 10, 38]. This type of methods relies on a sequence of decreasing $\lambda$ values and "warm start" (starting the active set with the solution from the previous $\lambda$) to achieve computational efficiency. Usually these methods have multiple iteration loops to incorporate the strong rule screening, active set, and path-wise coordinate descent. The inner loop performs coordinate descent and active set management. The outer-loop goes through a sequence of decreasing $\lambda$ values and initializes the active set at each $\lambda$ with the strong rule and warm start. Since they do not utilize safe convergence stopping criteria for the active set, they may miss some of the active features in the optimal solutions to the original LASSO formulation with the corresponding $\lambda$ values.

## 1.3 Working Set Method

Working set methods [15, 16, 26, 19, 20] maintain a working set according to some violation rules and solve a sub-problem regarding the working set at each step. [15] estimate an extreme feasible point based on the current solution, this method constructs the working set for the next step by the constraints that are closest to the feasible point. The method in [19, 20] employs dual extrapolation [29] strategies to improve the accuracy of dual variable estimation. With a more accurate dual variable, their method can improve feature recruiting and can exit early especially when the required solution accuracy is not high. While not effectively leveraging the power of feature screening during algorithm updating, this approach may recruit a large number of inactive features especially when a highly accurate solution is needed. Furthermore, working sets [15, 19, 20] usually try to solve a sequence of sub-problems with a high precision, thus it may further introduce redundant computational cost to the original problem.

## 1.4 Our Contributions

In this paper, we propose a novel safe LASSO feature selection method (Thunder [1]) to further scale up LASSO solutions by overcoming the issues in the existing methods. Our algorithm starts from a small set of features, which is taken as the active set $\mathcal{A}$. Time-consuming operations are performed on $\mathcal{A}$. The rest coordinates or features are stored in the remaining set $\mathcal{R}$. Features are actively recruited by or removed from the active set according to the operation rules derived from the estimation range of optimal dual variables. Based on the duality properties, safe stopping criteria have been developed to keep most inactive and redundant features out of the active set. The efficiency of the proposed approach is further improved by employing two remaining sets. Feature recruiting complexity can be relieved in addition to the reductions of inactive features thanks to both remaining sets.

The proposed Thunder solver is not a typical working set method that usually has to solve a sequence of sub-problems with high solution precision in order to approach the optimal solution. Thunder actively refines the active set by integrating feature screening and feature recruiting to maximally reduce the number of inactive features involved in the active set. The methods [15, 19, 20] have to sequentially solve a large number of sub-problems. Each sub-problem is required to reach the same solution precision asked by the user. The working sets of these sub-problem usually involve a large number of inactive features. Numerous redundant operations consumed by the inactive features involved in the sub-problems can harm the efficiency of the working set methods.

Meanwhile, Thunder can avoid a large number of inactive features by utilizing the derived safe stopping rule for feature recruiting. Once the stopping condition is reached, the updating steps of Thunder can produce a solution with any level of precision without recruiting any new features to the active set. While for some existing working set solvers [19, 20], when the given accuracy is not reached, they may double the working set size in each step of the outer loop without checking any stopping condition. The large number of inactive features taken by these solvers will reduce the algorithm efficiency especially when the required precision is high. With the safe stopping condition, Thunder can avoid these issues. Theoretical analysis shows that under high solution precision requests, the proposed Thunder complexity only relates to the number of true active features that the algorithm trying to recover. Experiments on both simulated and real-world data sets validate the advantages.

## 2 Methodology

Let $f^*$ be the conjugate of $f$ in problem (1), and the dual form [3, 23] is as follows,

$$D : \sup_\theta -\sum_{j=1}^n f^*(-\lambda\theta_j, y_j) \quad s.t. \quad |x_i^\top \theta| \leq 1, \quad \forall x_i \in \mathcal{F}. \tag{2}$$

Here $\theta$ is the dual variable, $\mathcal{F}$ is the feature set, and $x_i$ is the feature $i$ (i.e., $i$-th column of $X$). The optimal primal $\beta^*$ and the optimal dual variable $\theta^*$ relationship is $f'(\mathbf{x}_{j\bullet}\beta^*) = -\lambda\theta_j^*$, where $f'$ is the first-order derivative of $f$. Gap dynamic screening [21, 9, 2] can scale up sparse model solutions by dual variable range estimation during the algorithm iterations. Let $P(\beta)$ and $D(\theta)$ be the primal and dual objective value at $\beta$ and $\theta$, respectively. The ball region for $\theta^*$ is estimated based on the duality gap as a function of the primal and dual objective function values at iterative updates [21, 9]:

$$\forall\theta \in \Delta_\mathcal{F}, \beta \in R^{p\times 1}, \ B\Big(\theta, \frac{2}{\lambda^2}[P(\beta) - D(\theta)]\Big) = \Big\{\theta^* | ||\theta^* - \theta||_2^2 \leq \frac{2}{\lambda^2}[P(\beta) - D(\theta)]\Big\}. \tag{3}$$

Here $\Delta_\mathcal{F} = \{\theta \mid |x_i^\top \theta| \leq 1, \forall x_i \in \mathcal{F}\}$ is the dual feasible space corresponding to the feature set $\mathcal{F}$; $\beta$ is the current estimation of primal variables; and $\theta$ is the projected feasible dual variables of $\beta$. The proposed active incremental approach starts from a small active set $\mathcal{A}$ and updates the solutions of the corresponding sub-problem.

### 2.1 Solving the Sub-problem with an Active Set

At step $t$ (where $t$ is the index for the outer loop of the proposed algorithm), we use $\mathcal{A}_t$ to represent the active set, and $\mathcal{R}_t$ the remaining set ($\mathcal{F} = \mathcal{A}_t \cup \mathcal{R}_t$), respectively. The proposed active incremental method tries to solve the following sub-problem that focuses on $\mathcal{A}_t$.

$$P_t : \min_{\beta \in R^{|\mathcal{A}_t|}} \sum_{j=1}^n f\Big(\sum_{i:x_i \in \mathcal{A}_t} x_{ji}\beta_i, y_j\Big) + \lambda||\beta||_1. \tag{4}$$

$$D_t : \sup_\theta -\sum_{j=1}^n f^*(-\lambda\theta_j, y_j) \quad s.t. \quad |x_i^\top \theta| \leq 1, \quad \forall x_i \in \mathcal{A}_t, \tag{5}$$

Similar to gap dynamic screening [21, 9, 2], our approach relies on the dual variable estimation to actively recruit and remove features. We use $\bar{\mathcal{A}}$ to represent the optimal active feature set $\{x_i : |x_i^\top \theta^*| = 1\}$. Figure 1 shows the relations among $\mathcal{F}$, $\mathcal{A}_t$, and $\bar{\mathcal{A}}$. Based on the current estimated solution of the sub-problem, we define *feature recruiting* and *feature screening* operations to manipulate features between $\mathcal{A}_t$ and $\mathcal{R}_t$. Feature recruiting operation choose the potential active features in $\mathcal{R}_t$ and move them to $\mathcal{A}_t$.

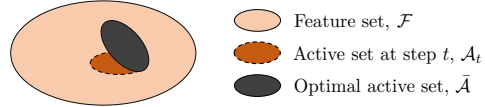

- Feature set, $\mathcal{F}$
- Active set at step $t$, $\mathcal{A}_t$
- Optimal active set, $\bar{\mathcal{A}}$

Figure 1: $\mathcal{A}_t$ starts from a small initial set, actively recruits features from $\mathcal{F} \cap \mathcal{A}_t^c$, and removes features unlikely belonging to $\bar{\mathcal{A}}$ according to the derived rules. $\mathcal{A}_t$ finally converges to set $\bar{\mathcal{A}}$.

According the to dual form of the problem (5), the activity of a feature $x_i$ is determined by the value $|x_i^\top \theta_t^*|$. The range of $\theta_t^*$ is estimated with ball region $||\theta_t - \theta_t^*|| \leq r_t$. $r_t$ is the radius of the ball region estimated with the duality gap according to (3). Let $\theta_t^*$ represent the optimal dual variable regarding the sub-problem at step $t$, we have the following lemma to show the relationship between the sub-problem and the original problem.

**Lemma 1** *With* $||\theta_t - \theta_t^*|| \leq r_t$ *as the dual estimation at step* $t$, *if* $\max_{i:x_i \in \mathcal{R}_t} |x_i^\top \theta_t| + ||x_i||_2 r_t < 1$, *then* $\theta_t^* = \theta^*$, *and we can safely stop the feature recruiting operations.*

**Proof**: *With* $\mathcal{A}_t \subseteq \mathcal{F}$, *we have* $\Delta_\mathcal{F} \subseteq \Delta_{\mathcal{A}_t}$, *and* $D(\theta^*) \leq D(\theta_t^*)$. *As* $\forall x_i \in \{\mathcal{R}_t = \mathcal{F} \setminus \mathcal{A}_t\}$, *we have* $|x_i^\top \theta_t^*| \leq |x_i^\top \theta_t| + ||x_i||_2 r_t < 1$, *and* $\theta_t^* \in \Delta_\mathcal{F}$. *With* $\theta^* = \sup_{\theta \in \Delta_\mathcal{F}} D(\theta)$, *we get* $D(\theta^*) \geq D(\theta_t^*)$. *As we already know* $D(\theta^*) \leq D(\theta_t^*)$, *we get* $D(\theta^*) = D(\theta_t^*)$. *Since the dual problem is concave and smooth, and the feasible set is closed and convex, it means* $\theta_t^* = \theta^*$. *It tells us that the active set already obtained all the features in the optimal feature set, i.e.,* $\bar{\mathcal{A}} \subseteq \mathcal{A}_t$, *and the feature recruiting operation can be stopped.* $\square$

Figure 2 gives an example to illustrate the above lemma. For a highly sparse data set, most trivial features can be avoided in the computation with an accurate dual estimation according to Lemma 1.

**Remark 1** *At step $t$, if a new feature is added into $\mathcal{A}_t$, then $\mathcal{A}_t \subseteq \mathcal{A}_{t+1}$, $\Delta_{\mathcal{A}_t} \supseteq \Delta_{\mathcal{A}_{t+1}}$, and $D(\theta^*_{t+1}) \leq D(\theta^*_t)$.*

Remark 1 shows that the algorithm can always converge with feature recruiting. The feature screening operation basically follows the gap screening rule [21]. The safety of screening and recruiting rules ensures the safety of the algorithm. Lemma 1 provides a safe stopping condition for the feature recruiting operation in the Thunder algorithm.

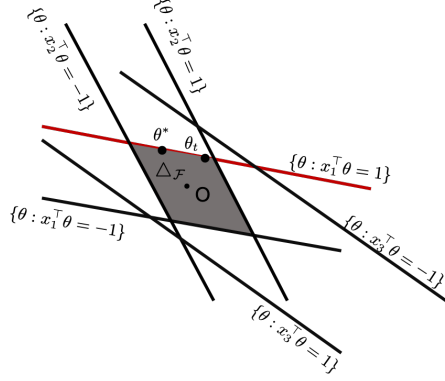

Figure 2: In this example, we have 3 features, $\{x_1, x_2, x_3\}$, and $\bar{\mathcal{A}} = \{x_1\}$. At step $t$, $\mathcal{A}_t = \{x_1, x_2\}$, $\mathcal{R}_t = \{x_3\}$, and $|x_3^\top \theta_t| + ||x_3||_2 r_t < 1$, then we do not need to move $x_3$ from $\mathcal{R}_t$ to $\mathcal{A}_t$.

**Feature Screening**: *For $x_i \in \mathcal{A}_t$, with $||\theta_t - \theta^*_t|| \leq r_t$, if $|x_i^\top \theta_t| + ||x_i||_2 r_t < 1$, move $x_i$ from $\mathcal{A}_t$ to $\mathcal{R}_t$.*

We will present more details about feature recruiting in the following two subsections.

## 2.2 Feature Recruiting with Sampling Strategy

For feature $x_i$, its activity can be estimated from the ball region estimation for $\theta^*_t$ ($||\theta_t - \theta^*_t|| \leq r_t$) as $|x_i^\top \theta_t| - ||x_i||_2 r_t \leq |x_i^\top \theta^*_t| \leq |x_i^\top \theta_t| + ||x_i||_2 r_t$. Based on the upper and lower bounds of feature activities, we define the following operation to move the potential active feature from $\mathcal{R}_t$ to $\mathcal{A}_t$.

**Feature Recruiting**: *For $x_i \in \mathcal{R}_t$, if $\forall \hat{i} : x_{\hat{i}} \in \mathcal{R}_t, \hat{i} \neq i$, $\left| |x_i^\top \theta_t| - ||x_i||_2 r_t \right| > |x_{\hat{i}}^\top \theta_t| + ||x_{\hat{i}}||_2 r_t$, move $x_i$ to $\mathcal{A}_t$.*

The cost of recruiting operations may dilute the benefits of the whole algorithm. Instead of checking the activity of features one by one, we jointly check a batch of features' activity in an approximation way. Let's use $\mathcal{H}$ to represent the top $\mu$ features based on the descending order of $|x_i^\top \theta_t|$. Since the recruiting operation is to select the most active features in $\mathcal{R}_t$, we use the least active one in $\mathcal{H}$ to decide whether we need to add $\mathcal{H}$ to the active set $\mathcal{A}_t$ or not. Rather than comparing $\mathcal{H}$ with all the features in $\mathcal{R}_t \setminus \mathcal{H}$, we only check with a small number of randomly sampled features from $\mathcal{R}_t \setminus \mathcal{H}$. This approach can significantly reduce the cost of feature selection. The algorithm for the recruiting operation is given in Algorithm 1, where we use a subset of $\mathcal{R}_t$ to determine whether or not we accept the recruited features.

---
**Input:** $\theta_t, r_t, \mathcal{R}_t, \mathcal{A}_t, X, \tau$ $(0 < \tau < 1)$
**Result:** $\mathcal{R}_{t+1}, \mathcal{A}_{t+1}$

---
$\mu \leftarrow \lceil |\mathcal{A}_t|/2 \rceil$
//Select recruiting candidates:
Let $\mathcal{H}$ be the subset of $\mathcal{R}_t$ containing the elements with the first $\mu$ largest values of $|x_i^\top \theta_t|$;
//Sample a subset to compare with:
Construct a set $\mathcal{B}$ with randomly selected elements in set $\mathcal{R}_t \setminus \mathcal{H}$;
$i \leftarrow \min_{\hat{i}:x_{\hat{i}} \in \mathcal{H}} |x_i^\top \theta_t|$;
$\mathcal{V} \leftarrow \{x_{\hat{i}} | x_{\hat{i}} \in \mathcal{B}, |x_i^\top \theta_t| - ||x_i||_2 r_t \leq |x_{\hat{i}}^\top \theta_t| + ||x_{\hat{i}}||_2 r_t\}$;
//Accept or reject:
**if** $|\mathcal{V}|/|\mathcal{B}| < \tau$ **then**
  $\quad \mathcal{A}_{t+1} \leftarrow \mathcal{A}_t \cup \mathcal{H}$;
  $\quad \mathcal{R}_{t+1} \leftarrow \mathcal{R}_t \setminus \mathcal{H}$;
**end**

---
**Algorithm 1:** Feature Recruiting

## 2.3 Improve Feature Recruiting with Bi-level Selection

To further improve the efficiency of feature recruiting, we split the remaining set to two sub-remaining-sets $\mathcal{R}^1$ and $\mathcal{R}^2$, i.e., $\mathcal{R} = \mathcal{R}^1 \bigcup \mathcal{R}^2$. The first remaining set $\mathcal{R}_t^1$, which is closer to the active set $\mathcal{A}_t$, stores the features with relatively higher activity compared with the features in the second one, $\mathcal{R}_t^2$. The algorithm actively chooses the features with high activity in $\mathcal{R}_t^1$ and move them to $\mathcal{A}_t$ with the feature recruiting operations. Features can be moved between $\mathcal{R}_t^1$ and $\mathcal{R}_t^2$ based on their activity with the shrinking operations. With $\varsigma$ as the size ratio between $\mathcal{R}_t^1$ and $\mathcal{R}_t^2$, the shrinking operation is given as follows:

**Feature Shrinking**: *Sorting all the features in $\mathcal{R}_t = \{x_i | x_i \in \mathcal{R}_t^1 \cup \mathcal{R}_t^2\}$ according to the descending order of $|x_i^\top \theta_t|$, and take the first $\varsigma|\mathcal{R}_t|$ features as the $\mathcal{R}_{t+1}^1$, and the rest $(1 - \varsigma)|\mathcal{R}_t|$ ones as $\mathcal{R}_{t+1}^2$.*

With the three defined operations, the algorithm tries to keep the most active features in $\mathcal{A}_t$, the features with high potential activity in $\mathcal{R}_t^1$, and potentially inactive features in $\mathcal{R}_t^2$. Figure 3 shows the scheme of the proposed approach. Algorithm 2 gives the flow of the proposed Thunder algorithm. We can see the algorithm will reassign features in the remaining sets to $\mathcal{R}_t^1$ and $\mathcal{R}_t^2$ every $K_2$ outer iterations.

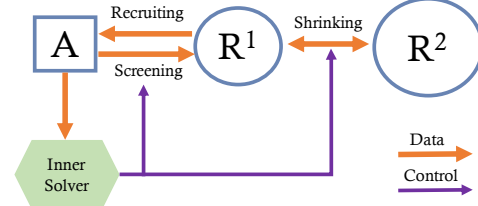

Figure 3: The scheme of the proposed method.

---

**Input:** $X, \mathbf{y}, \lambda, \epsilon$
**Result:** $\beta$

---

Choose a small set of features from $\mathcal{F}$ in the descending order of $|X^\top \mathbf{f}'(\mathbf{0})|$;
*DoRecruit* $\leftarrow$ **True**, $t \leftarrow 0$;
**while** *True* **do**
    Update $\beta_t$ with $K_1$ iterations of soft-thresholding operations on $\mathcal{A}_t$; //Sub-problem solver
    Compute a ball region $B(\theta_t, r_t)$; //Dual range estimation
    **if** *DoRecruit = False & Duality Gap < $\epsilon$* **then**
        | **Stop**; //Algorithm exits
    **end**
    Feature Screening; $t \leftarrow t + 1$;
    **if** $t \mod K_2 = 0$ **then**
        | Feature Shrinking;
    **end**
    **if** *DoRecruit = False* **then**
        | **Continue**; //Recruiting already stopped
    **else**
        | //Determine stop recruiting or not (Lemma 1):
        | **if** $\max_{x_i \in \mathcal{R}_t^1} |x_i^\top \theta_t| + ||x_i||_2 r_t < 1$ **and** $\max_{x_i \in \mathcal{R}_t^2} |x_i^\top \theta_t| + ||x_i||_2 r_t < 1$ **then**
        | | *DoRecruit* $\leftarrow$ **False**; **Continue**;
        | **end**
        | Feature Recruiting; //If decide to continue with recruiting
    **end**
**end**
Put $\beta_t$ into $\beta$, and set the other entries to be 0.

**Algorithm 2:** Thunder Algorithm

---

During the updates, the inter-products between inactive features and the dual variable will finally converge to values in [-1, 1]. The next lemma illustrates the shrinking property [14] of the features.

**Lemma 2** *Let $\beta^*$ be the optimal solution of the primal problem (1), and $\theta_t$ be the current estimation of the dual variable for (2), then a) If $\beta_i^* = 0$, then $\exists t_i, \forall t > t_i, |x_i^\top \theta_t| \leq 1$; b) If $\beta_i^* \neq 0$, then $\exists t_i$ and $\epsilon, \forall t > t_i, \left| |x_i^\top \theta_t| - 1 \right| < \epsilon$.*

**Proof**: *According to the KKT condition of the primal problem (1), at the optimal point, we have $x_i^\top \theta^* \begin{cases} = sign([\beta^*]_i) & if [\beta^*]_i \neq 0 \\ \in [-1,1] & if [\beta^*]_i = 0. \end{cases}$ Then we have $\lim_{t \to \infty} |x_i^\top \theta_t| \leq 1$, if $\beta_i^* = 0$; $\lim_{t \to \infty} |x_i^\top \theta_t| = 1$ if $\beta_i^* \neq 0$. Thus if $\beta_i^* = 0$, then $\exists t_i, \forall t > t_i, |x_i^\top \theta_t| \leq 1$, and if $\beta_i^* \neq 0$, then $\exists t_i$ and $\epsilon, \forall t > t_i, \left| |x_i^\top \theta_t| - 1 \right| < \epsilon$.* $\square$

Lemma 2 tells us that, with a larger $t$, the algorithm is more confident about the activities of the features. For an inactive feature $x_i$, the value $\lim_{t \to \infty} |x_i^\top \theta_t| \leq 1$. With the second redundant set $\mathcal{R}_t^2$, we try to reduce the number of inactive features involved in the recruiting operations by leveraging the shrinking strategy [14]. We can store the less active features in the second remaining set $\mathcal{R}_t^2$. In each outer iteration, the recruiting operation only needs to consider the activity of the features in $\mathcal{R}_t^1$, rather than the whole feature set outside of $\mathcal{A}_t$.

This approach reduces the computation cost for recruiting operations. To ensure the safety of the algorithm, we just need to check the activity of the features in $\mathcal{R}_t^2$ every $K_2$ outer iterations, and reassign the membership in both $\mathcal{R}_t^1$ and $\mathcal{R}_t^2$. The steps for shrinking operation are described in Algorithm 3. The accuracy of the dual variable estimation is essential for the defined three operations. In our implementation, we employ the extrapolation strategy [19, 29] to improve dual estimation.

---
**Input:** $\mathcal{R}_t^1, \mathcal{R}_t^2, \theta_t, \varsigma$
**Result:** $\mathcal{R}_{t+1}^1, \mathcal{R}_{t+1}^2$

---
$\mathcal{R}_t \leftarrow \mathcal{R}_t^1 \bigcup \mathcal{R}_t^2$
$\sigma \leftarrow \lceil \varsigma |\mathcal{R}_t| \rceil^{th}$ largest value in set $\{|x_i^\top \theta_t| \mid x_i \in \mathcal{R}_t\}$
$\delta \leftarrow \max(\sigma, 1);$
$\mathcal{R}_{t+1}^1 \leftarrow \{x_i | |x_i^\top \theta_t| \geq \delta; x_i \in \mathcal{R}_t\}$
$\mathcal{R}_{t+1}^2 \leftarrow \{x_i | |x_i^\top \theta_t| < \delta; x_i \in \mathcal{R}_t\}$

---
**Algorithm 3:** Feature Shrinking

## 3 Convergence Analysis

The sub-problem solver in Algorithm 2 can be ISTA/FISTA [1], or other coordinate descent methods. Our algorithm employs a cyclic block coordinate minimization/descent (CM/CD) type method [11] as the sub-problem solver. Different from working set methods [15, 19], the inner solver in Thunder is not required to converge to update the active set. After $K_1$ coordinate minimization or descent steps with the inner solver, we do screening and recruiting to update the active set $\mathcal{A}_t$. The shrinking operation is performed every $K_2$ outer iterations.

### 3.1 Complexity Analysis for Feature Recruiting

The solution accuracy of the dual problem is almost linearly proportional to the accuracy of the primal problem after a number of algorithm iterations. Let $\mathbf{f}(X\beta) = \sum_{j=1}^n f(x_{j\bullet}\beta, y_j)$, and we further assume that $\mathbf{f}$ is $\gamma$-convex function in our analysis. For nonstrongly convex minimization, we only need to add a strongly convex perturbation to the objective function to meet the assumption [17, 30]. Let $\bar{L} = \sqrt{\sigma_{\max}}L$, $\sigma_{\max}$ is the largest eigenvalue of $X^\top X$, $L$ is the Lipschitz constant of the gradient of $\mathbf{f}$. Similarly, $\bar{L}_t$ is the Lipschitz constant of the gradient of the sub-problem objective function at step $t$ [17]. The following lemma tells us that, with more features, the Lipschitz constant of the loss function's gradient will also increase.

**Lemma 3** *With one recruiting operation at step t, we have $\bar{L}_{t+1} \geq \bar{L}_t$. For all the sub-problems, we have $\bar{L} \geq \bar{L}_t$.*

**Proof:** *With the recruiting operation, we can see $\mathcal{A}_t \subseteq \mathcal{A}_{t+1}$. Without losing generality, we add one feature $u$ to $X_t$, i.e., $X_{t+1} = [X_t, u]$. Since $\sigma_{max}^{t+1} = \sup_{||v||=1} \left\| v^\top X_{t+1}^\top X_{t+1} v \right\| \geq \left\| \begin{bmatrix} v_t \\ 0 \end{bmatrix}^\top [X_t, u]^\top [X_t, u] \begin{bmatrix} v_t \\ 0 \end{bmatrix} \right\| = \sigma_{max}^t$, then we get $\bar{L}_{t+1} \geq \bar{L}_t$, and $\bar{L} \geq \bar{L}_t, \forall t$.* $\square$

A larger Lipschitz constant usually requires more steps for the algorithm to converge. A sub-problem with more features means more computation steps. The algorithm has two main phases, feature recruiting and feature screening. Feature recruiting corresponds to the iterations when DoRecruit is True; feature screening phase corresponds to the iterations after DoRecruit is changed to False. Let $H$ be the total number of features involved in the recruiting operation; after the $h$-th feature ($h$ in the sequence of $\{1, 2, ..., h, ..., H\}$) has been added into the active set, we use $P_h$, $p_h$, and $\beta_h^*$ to denote the primal objective function, the size of the active set, and the optimal primal solution of the sub-problem, respectively. Let $Q_h(\beta) = P_h(\beta) - P_h(\beta_h^*)$ be the primal objective accuracy to adding feature $h$. With $O(u)$ as the complexity for one iteration of coordinate minimization, the following lemma gives the complexity for the feature recruiting phase.

**Lemma 4** *With $O(u)$ as the complexity for one iteration of coordinate minimization for a LASSO-type problem, $H$ is the total number of features involved in recruiting operations, and $p_H$ the size of the active set when DoRecruit is set to false, the complexity for the feature recruiting phase of the proposed algorithm is $O\left( \left(u + p\left(\frac{n\varsigma}{K_1} + \frac{n(1-\varsigma)}{K_1 K_2}\right)\right)\left(\mathcal{U} + \frac{\bar{L}^2}{\gamma^2}\Phi + p_H \frac{\bar{L}^2}{\gamma^2} \log \frac{\bar{Q}}{Q_H(\beta_H)}\right) \right)$, where $\bar{Q} = \left(\Pi_{h=1}^{H-1} Q_{h+1}(\beta_h)^{p_{h+1}-p_h}\right)^{\frac{1}{p_H}}$, $\mathcal{U} = \log\left(\Pi_{h=1}^{H-1} \frac{Q_{h+1}(\beta_h)^{\frac{p_h}{p_{h+1}}}}{Q_h(\beta_h)^{\frac{p_h}{p_{h+1}}}} \frac{1}{Q_H(\beta_H)}\right)$, and $\Phi = \log\left(\Pi_{h=1}^{H-1} \frac{Q_{h+1}(\beta_d)^{p_h}}{Q_h(\beta_h)^{p_h}}\right)$.*

A larger $K_1$ may lead to a smaller value in the first part of the complexity, but it may increase the second part of it. It is due to that a larger $K_1$ can increase the primal precision $Q_h(\beta) = P_h(\beta) - P_h(\beta_h^*)$ for feature $h$ to be added to the active set $\mathcal{A}_t$, and redundant operations will be introduced to pursue an unnecessary higher precision for a sub-problem (4) at step $t$. We also notice that the accuracy for the last recruited feature ($Q_H(\beta_H)$) is important. However, the $H$-th feature might not be in the optimal set $\bar{\mathcal{A}}$. Early stop of feature recruiting is critical as it can reduce the number of inactive features involved in recruiting phase. It also reduces the complexity of feature screening.

## 3.2 Complexity of Thunder Algorithm

The complexity of the proposed method is given by the following theorem.

**Theorem 1** *With $O(u)$ as the complexity for one iteration of coordinate minimization of the LASSO problem with a $\gamma$-convex loss function, the time complexity for the proposed algorithm is $O\left(u\frac{\bar{L}^2}{\gamma^2}\left(\eta\bar{p}\log\frac{\bar{Q}}{\varepsilon_D} + \eta\bar{p}H + |\bar{\mathcal{A}}|\log\frac{\varepsilon_D}{\varepsilon}\right)\right)$. Here $H$ is the total number of features involved in recruiting operations, $\bar{p}$ is the maximum size of the active set during the algorithm iterations, $\bar{Q}$ is the geometric mean of the sub-problem primal objective function precision values corresponding to each recruiting operation, and $\varepsilon_D$ is the primal objective function precision for the last feature screening operation. $\eta = 1 + \frac{np\varsigma}{uK_1} + \frac{np(1-\varsigma)+p\log p}{uK_1 K_2}$, and $\varsigma$ is the feature partition ratio for $\mathcal{R}_1$ and $\mathcal{R}_2$.*

**Remark 2** *From Theorem 1, we can see that the complexity of the proposed method is determined by both $\varepsilon_D$ and $\varepsilon$. Under the assumption $\varepsilon < \varepsilon_D$, the requested accuracy will affect the whole algorithm complexity with coefficient $|\bar{\mathcal{A}}|$.*

Besides the feature number ($p$) and the sample number ($n$), Theorem 1 shows that the optimal active set size ($|\bar{\mathcal{A}}|$), the number of features involved in recruiting operations ($H$), and the maximum size of the active set ($\bar{p}$) impact the algorithm efficiency as well. The less number of trivial features recruited by $\mathcal{A}$, the more efficiency the algorithm can achieve. Theorem 1 also indicates that the value of $K_1$ can be set proportional to the product of $n$ and $p$. After reaching the precision $\epsilon_D$, the complexity for Thunder to reach $\epsilon$ is only proportional to $|\bar{\mathcal{A}}|$. With the safe recruiting stopping condition based on Lemma 1, Thunder can avoid many trivial features in pursuing high solution accuracy. Without the safe stopping condition for feature adding, the large number of trivial features will impair the efficiency of most working set methods [15, 19] to reach high precision solutions. The correlation between features may affect the efficiency of Thunder, but it does not impact the algorithm's safety. Based on the proof of Theorem 1, the optimal $K_1$ can be a value proportional to $\sqrt{np/u}$.

## 4 Experiments

In this section, we present experiments to compare the proposed method with other existing sparse optimization methods. We evaluate the selected methods for the LASSO formulation with a simulation data set and three real-world data sets. We specifically focus on the performance comparison among (1) working set method [15] (Blitz1), (2) the recently proposed Celer [19]. We use the online packages for the BLITZ [15] and Celer [19] methods, respectively. For Celer, experimental results are based on the initial version published with paper [19]. All the three algorithms are safe methods that do not require the help from a heavier penalty parameter, and they are implemented with Cython or C/C++ wrapped with python. We set $\varsigma = \frac{1}{3}$ and the initial size of $\mathcal{A}$ as 50 for Thunder in the following experiments. We use $\lambda_{max}$ to represent the smallest $\lambda$ value that leads to all zero $\beta^*$ entries.

### 4.1 Simulation Study

First, we simulate the data sets with $n = 10^4$ samples and $p = 10^8$ features according to a linear model $\mathbf{y} = X\beta + \epsilon$, where each column of $X$ is a vector with random values uniformly sampled from the interval $[-1.0, 1.0]$, and the white noise $\epsilon \sim N(0, 0.1)$. For the linear coefficients $\beta$, 10% entries ($0.1p$) are randomly set to the values in $[1.0, -1.0]$, and the rest ($0.9p$) to zero. For this data set, we can derive $\lambda_{max} = 31.41$. The first plot in Figure 4 illustrates the running time of different methods with $\lambda = 0.3\lambda_{max}, 0.5\lambda_{max}$, and $0.7\lambda_{max}$ at stopping accuracy $1.0\mathrm{E}-4$. The third plot gives the results at duality gap $1.0\mathrm{E}-7$. The second and fourth plots show the running time ratio at two duality gaps, respectively. We can see that the proposed method is much faster compared to other methods in reaching the optimal solution under a specified accuracy. The results also show that Thunder is more efficient compared with the existing safe methods when $\lambda$ is small.

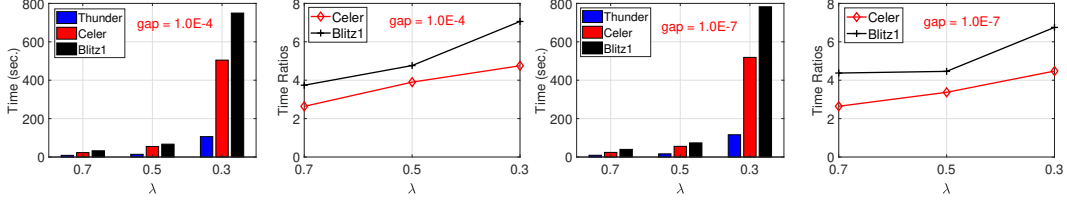

Figure 4: **Simulated data**. Running times of three algorithms at three $\lambda$ values, for two duality gap values ($1.0\mathrm{E}-4$ and $1.0\mathrm{E}-7$). Note that the $x$-axis is $\lambda/\lambda_{max}$. The 1st and 3rd plots depict the absolute execution times while the 2nd and 4th plots show the improvement of efficiency for Thunder.

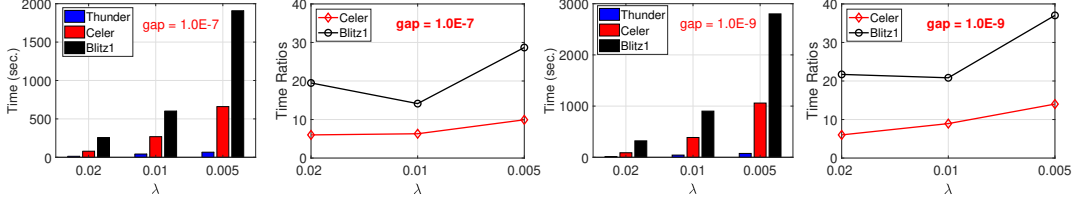

Figure 5: **Finance data**. Running times of three algorithms at three $\lambda$ values, for two duality gap values ($1.0\mathrm{E}-7$ and $1.0\mathrm{E}-9$). Note that the $x$-axis is $\lambda/\lambda_{max}$. The 1st and 3rd plots depict the absolute execution times while the 2nd and 4th plots show the improvement of efficiency for Thunder.

## 4.2 Finance Data Set

The finance data set (E2006-log1p) is publicly available on LIBSVM website. After pre-processing with the methods in [19], there are 16,087 samples, and 1, 668, 738 features in the data set. The first plot in Figure 5 gives the running times of different solvers at the duality gap $1.0\mathrm{E}-7$ with $\lambda$ values $0.02\lambda_{max}$, $0.01\lambda_{max}$ and $0.005\lambda_{max}$. The third plot presents running times at the duality gap $1.0\mathrm{E}-9$. The second and fourth plots present running time ratio at two duality gaps, respectively. The proposed Thunder algorithm takes least time in both cases.

The results in this set of experiments show that higher solution accuracy does not lead to drastic computation increase for the proposed method. To ensure this conclusion, we further evaluate the proposed method and Celer [19] with $\lambda$ fixed to $0.005\lambda_{max}$ and solution accuracy varying from 1.0E-4 to 1.0E-11. From the results in Figure 6, we can see that with the pre-specified accuracy $\epsilon$ decreasing, the running time consumed by Celer increases dramatically. While the computation increment for the proposed method is relatively marginal. With Figure 7, we further investigate the active set

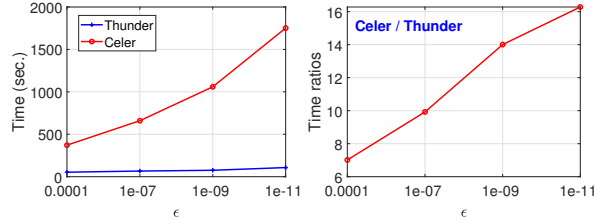

Figure 6: The left column is the running time of Thunder and Celer at different solution accuracy with fixed $\lambda$ on the finance data set. Besides, the right column illustrates the time ratios between Celer and Thunder.

size of Thunder and the working set size of Celer during algorithm updating with $\lambda = 0.005\lambda_{max}$. We can see that during the updating steps of Thunder the active set size is always around $|\bar{\mathcal{A}}|$. While for Celer, the working set size is always increasing and this may hurt its performance. The experimental results are consistent with the theoretical analysis in the previous section.

## 4.3 LASSO Path

In practice, people usually solve LASSO problems involve a sequence of $\lambda$ values to choose the best one. Given a sequence of decreasing $\lambda$ values, we adapt Thunder to LASSO path problems with warm starting strategy, i.e., initializing the active set $\mathcal{A}$ with the solution from a larger $\lambda$ value. We compare Thunder with Celer on a coarse LASSO path problem for both the KDD2010 and URL data sets (more details about the data sets in supplemental file). We evenly select 10 $\lambda$ values on the logarithmic scale of the range $[\lambda_{max}, 0.01\lambda_{max}]$. Figure 8 gives the running time for both solvers at duality gaps 1.0E-5 and 1.0E-8. The results show that the Thunder takes much less computation for

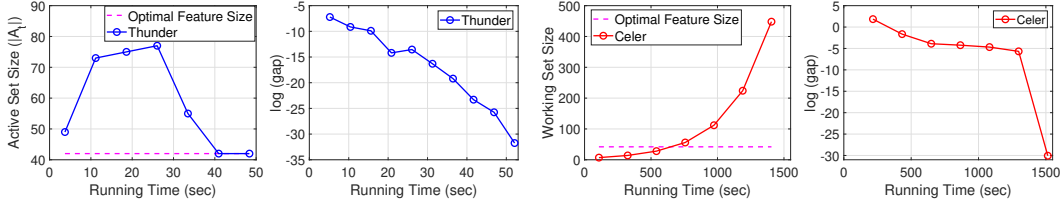

Figure 7: The 1st plot shows the active set size of Thunder, $|\mathcal{A}|$, the 2nd one gives the duality gap of Thunder. The 3rd plot is the working set size of Celer, and the last plot gives the duality gap of Celer.

both data sets at different accuracy levels. Similar to single $\lambda$ problems, Thunder has more advantages with higher solution accuracies. More experimental results can be found in the supplemental file.

## 5 Discussion

Thunder follows a passive feature adding manner compared to [19]. In each outer loop of Thunder, the inner solver runs a small number ($K_1$) of updating steps, and then the algorithm conducts feature screening and recruiting. Feature screening (GAP screening, [21]) is always safe regarding the current sub-problem and active set $\mathcal{A}$. Feature recruiting is performed passively under the condition given by line 6 in Algorithm 1. If the

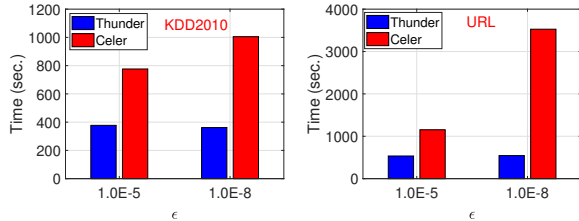

Figure 8: Running time of Thunder and Celer to solve LASSO path at different duality gaps.

recruiting condition is met, only a small number of features will be added to $\mathcal{A}$, otherwise do nothing. The advantages for passive feature adding are:

- The passive approach can ensure that the recruited features always have high potentiality in order to keep more redundant features out of the active set.

- Dynamical checking of the feature screening and recruiting can keep maximum redundant feature out of $\mathcal{A}$ in order to keep $\mathcal{A}$'s size mall.

- Most importantly, with small numbers of recruiting features, the passive feature recruiting method can ensure that the new sub-problems likely to have smaller initial duality gaps. Smaller initial duality gaps are important because they are essential to ensure the GAP feature screening to continuously maintain its screening power.

- Sampling strategies can be used to significantly reduce the complexity of condition checking in the feature recruiting.

These strategies can ensure that Thunder has a smaller active set during algorithm updating, effective feature screening, efficient feature recruiting. Moreover, the proposed safe feature recruiting stopping condition ensures Thunder can give solutions to any level of high precision with only very little computation addition. Thunder can potentially be extended to high dimensional large-data-sample problems with stochastic optimization [27, 36].

## 6 Conclusions

In this manuscript, we propose a new $\ell_1$ sparse learning solver. With the proposed approaches to improve the efficiency of coordinate selection, the computation cost of sparse learning can be further reduced. Our theoretical analysis shows that the computation cost for higher solution accuracy is only proportional to the optimal active features. Experiments on synthetic and real-world data sets show the advantages of the proposed method and validation of the analysis. Our future work includes further theoretical analysis of the algorithm complexity, and theoretical and experimental study of hyper-parameters, such as the sensitivity of the feature partition ratio $\varsigma$. The proposed algorithm can be extended to more general loss functions such as logistic regression and support vector machines.

## Broader Impact

Sparse learning methods, e.g., LASSO, have broad applications in real-world problems, such as price prediction, biological data analysis. Real-world data sets usually come with high dimensionality and involve too much unpredictable noise. LASSO is a fundamental statistic tool to select important features and improve the prediction as well. Moreover, with the simple form and theoretical guarantees already studied by many people, these types of models can provide us interpretation about the data and reliable prediction results as well. Algorithms proposed in this paper can scale up the solutions of sparse learning. The training procedure can be reduced even under high solution precision requests. Our method can potentially enlarge the application of sparse learning to scenarios such as real-time high dimensional data processing.

## Acknowledgments and Disclosure of Funding

We thank the anonymous Referees and Area Chair for their constructive comments. The work is supported by Baidu Research.

## Footnotes

[1]https://github.com/ShaogangRen/Thunder

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
