[Supplementary Material]

**Appendix-A. More Experimental Results**

**A-1. KDD2010 Data Set**

Figure 9: Computation time at different $\lambda$ values and duality gaps on KDD2010 data set. The last plot presents Celer and Thunder time-ratios.

KDD2010 is another large-scale data set from LIBSVM website. There are 8,407,752 data samples, and 20,216,830 features in the data set. The data set is from Carnegie Learning and DataShop, and was used in KDD Cup 2010. As BLITZ takes much more computation time than Thunder and Celer on large data sets, we do not include it in the results from this set of experiments. Figure 9 compares the running time for Thunder and Celer at different $\lambda$ values and solution accuracies. The solution accuracies are 1.0E-5, 1.0E-8, and 1.0E-11. The $\lambda$ values include $0.01\lambda_{max}$ and $0.001\lambda_{max}$. Figure 9 shows that the running time of Thunder increases slowly with the solution accuracy increasing, and the time used by Celer increases significantly.

**A-2. URL Data Set**

Figure 10: Running time of Celer and Thunder on URL data set at different $\lambda$ values and duality gaps.

We further compare Thunder with Celer on one more LIBSVM data set. There are 3,231,961 features and 2,396,130 samples in the URL data set. Similar to the finance and KDD2010 data sets, the data are stored in sparse format. We test both solvers at three $\lambda$ values, $0.01\lambda_{max}$, $0.005\lambda_{max}$, and $0.001\lambda_{max}$. Figure 10 gives the running times at two solution resolutions, 1.0E-5 and 1.0E-8. The plots illustrate that Thunder consistently takes much less time at different $\lambda$ values and duality gaps.

## A-3. More Results for LASSO Path on URL Data Set

We evenly select 50 $\lambda$ values on the logarithmic scale of the range $[\lambda_{max}, 0.001\lambda_{max}]$ for the URL data set. Table 1 gives the running for both methods.

| Method | Time (Sec.) |
|---|---|
| Celer | 61936.9 |
| Thunder | 15823.4 |

Table 1: Running time of LASSO path with Celer and Thunder on URL data set at accuracy 1.0E-8.

## Appendix-B. Convergence of Feature Recruiting

**Lemma 4** *With $O(u)$ as the complexity for one iteration of coordinate minimization for a LASSO-type problem, $H$ as the total number of features involved in recruiting operations, and $p_H$ as the size of the active set when DoRecruit is set to false, the complexity for the feature recruiting phase of the proposed algorithm is*

$$O\Bigg(\Big(u + p\big(\frac{n\varsigma}{K_1} + \frac{n(1-\varsigma)}{K_1 K_2}\big)\Big)\big(\mathcal{U} + \frac{\bar{L}^2}{\gamma^2}\Phi + p_H \frac{\bar{L}^2}{\gamma^2} \log \frac{\bar{Q}}{Q_H(\beta_H)}\big)\Bigg),$$

*where*

$$\bar{Q} = \big(\Pi_{h=1}^{H-1} Q_{h+1}(\beta_h)^{p_{h+1}-p_h}\big)^{\frac{1}{p_H}},$$

$$\mathcal{U} = \log\big(\Pi_{h=1}^{H-1} \frac{Q_{h+1}(\beta_h)}{Q_h(\beta_h)^{\frac{p_h}{p_{h+1}}}} \frac{1}{Q_H(\beta_H)}\big),$$

*and*

$$\Phi = \log\big(\Pi_{h=1}^{H-1} \frac{Q_{h+1}(\beta_d)^{p_h}}{Q_h(\beta_h)^{p_h}}\big).$$

**Proof**: *To prove Lemma 4, we assume there is an accuracy threshold for each feature to be added into the active set, and then we just need to add up all the operations required to reach the thresholds for all the final active features. Let $Q_h(\beta) = P_h(\beta) - P_h(\beta_h^*)$. According to the coordinate minimization analysis in [17], the updating iteration number to add the $h$-th feature is approximately given by $\log_{\psi_h} \frac{Q_h(\beta_h)}{Q_h(\beta_{h-1})}$, where $\psi_h = \frac{p_h \bar{L}_h^2}{p_h \bar{L}_h^2 + \gamma^2}$. We add up the time complexity of the outer loops regarding each added feature. $O(K_1 u)$ is the time complexity for $K_1$ base CM operations; $\nu_h$, $\varsigma\vartheta_h$ and $(1-\varsigma)\vartheta_h$ are the size of set $\mathcal{A}$, $\mathcal{R}^1$ and $\mathcal{R}^2$, respectively. $O(n(\nu_h + \varsigma\vartheta_h))$ is the computation complexity for duality gap and the recruiting operation in one iteration of outer loop. The complexity upper bound $T_a$ for feature recruiting phase is*

$$T_a \leq \sum_{h=1}^{H} \frac{\log_{\psi_h} \frac{Q_h(\beta_h)}{Q_h(\beta_{h-1})}}{K_1}(K_1 u + n\varsigma\vartheta_h) + \sum_{h=1}^{H} \frac{\log_{\psi_h} \frac{Q_h(\beta_h)}{Q_h(\beta_{h-1})}}{K_1 K_2}(n(1-\varsigma)\vartheta_h + \vartheta_h \log \vartheta_h)$$

$$\leq u \sum_{h=1}^{H} \log_{\psi_h} \frac{Q_h(\beta_h)}{Q_h(\beta_{h-1})} + \Big(\frac{n\varsigma}{K_1} + \frac{n(1-\varsigma) + \log p}{K_1 K_2}\Big)\sum_{h=1}^{H} \vartheta_h \log_{\psi_h} \frac{Q_h(\beta_h)}{Q_h(\beta_{h-1})}.$$

*Let $\phi = \frac{n\varsigma}{K_1} + \frac{n(1-\varsigma)+\log p}{K_1 K_2}$, then*

$$T_a \leq \sum_{h=1}^{H} \log_{\psi_h} \frac{Q_h(\beta_h)^{u+\phi\vartheta_h}}{Q_h(\beta_{h-1})^{u+\phi\vartheta_h}}.$$

*With $\Phi_h = u + \phi\vartheta_h$, we have*

$$T_a \leq \sum_{h=1}^{H} \log_{\psi_h} \frac{Q_h(\beta_h)^{\Phi_h}}{Q_h(\beta_{h-1})^{\Phi_h}}$$

$$= -\log_{\psi_1} Q_1(\beta_0)^{\Phi_1} + \sum_{h=1}^{H-1} \left( \log_{\psi_h} Q_h(\beta_h)^{\Phi_h} - \log_{\psi_{h+1}} Q_{h+1}(\beta_h)^{\Phi_{h+1}} \right) + \log_{\psi_H} Q_H(\beta_H)^{\Phi_H}$$

$$= \overbrace{\log_{\psi_H} Q_H(\beta_H)^{\Phi_H}}^{A} - \log_{\psi_1} Q_1(\beta_0)^{\Phi_1} + \overbrace{\sum_{h=1}^{H-1} \log_{\psi_{h+1}} \frac{Q_h(\beta_h)^{\frac{\log \psi_{h+1}}{\log \psi_h}\Phi_h}}{Q_{h+1}(\beta_h)^{\Phi_{h+1}}}}^{B} \,.$$

*With $\psi_h = \frac{p_h \bar{L}_h^2}{p_h \bar{L}_h^2 + \gamma^2}$, $k \geq \log^{-1}(\frac{k}{k-1})$, we have*

$$\log^{-1}\left(\frac{p_h \bar{L}_h^2 + \gamma^2}{p_h \bar{L}_h^2}\right) \leq \frac{p_{h+1}\bar{L}_{h+1}^2 + \gamma^2}{\gamma^2}.$$

*Each term in B becomes*

$$\log_{\psi_{h+1}} \frac{Q_h(\beta_h)^{\frac{\log \psi_{h+1}}{\log \psi_h}\Phi_h}}{Q_{h+1}(\beta_h)^{\Phi_{h+1}}} \leq \frac{p_{h+1}\bar{L}_{h+1}^2 + \gamma^2}{\gamma^2} \cdot \log \frac{Q_{h+1}(\beta_h)^{\Phi_{h+1}}}{Q_h(\beta_h)^{\frac{\log \psi_{h+1}}{\log \psi_h}\Phi_h}}.$$

*Let $m_h = \frac{\gamma^2}{p_h \bar{L}_h^2}$, $\bar{m}_h = \frac{1}{1 - \frac{m_h}{2}}$*

$$\frac{\log \psi_{h+1}}{\log \psi_h} = \frac{\log(1 + \frac{\gamma^2}{p_{h+1}\bar{L}_{h+1}^2})}{\log(1 + \frac{\gamma^2}{p_h \bar{L}_h^2})} = \frac{\sum_{i=1}^{\infty}(-1)^{i+1}m_{h+1}^i/i}{\sum_{i=1}^{\infty}(-1)^{i+1}m_h^i/i}$$

$$\leq \frac{m_{h+1}}{m_h - m_h^2/2} = \frac{m_{h+1}/m_h}{1 - m_h/2} = \frac{m_{h+1}}{m_h}\left(1 + \frac{m_h}{2}\bar{m}_h\right).$$

*Here $w = u + \phi p$, $M_{0h} = \frac{\bar{L}_h^2}{\gamma^2}$, $M_{1h} = \bar{m}_h + m_h$, $M_{2h} = \frac{\bar{L}_h^2}{\bar{L}_{h+1}^2 p_{h+1}} + M_{0h}$ As $\bar{L}_h \approx \bar{L}_{h+1}$, $\forall h$, without loss of the generality, we use $\bar{L}$ to replace all of the $\bar{L}_h$.*

*As $M_{1h} = \bar{m}_h + m_h$,*

$$\frac{p_{h+1}\bar{L}_{h+1}^2 + \gamma^2}{\gamma^2} \log \frac{Q_{h+1}(\beta_h)^{\Phi_{h+1}}}{Q_h(\beta_h)^{\frac{\log \psi_{h+1}}{\log \psi_h}\Phi_h}}$$

$$\leq \left(1 + \frac{p_{h+1}\bar{L}_{h+1}^2}{\gamma^2}\right) \log \frac{Q_{h+1}(\beta_h)^{u + \phi(p - p_{h+1})}}{Q_h(\beta_h)^{\frac{m_{h+1}}{m_h}\left(1 + \frac{m_h}{2}\bar{m}_h\right)(u + \phi(p - p_h))}}$$

$$= \left(u + p\phi + ((u + p\phi)\frac{\bar{L}_{h+1}^2}{\gamma^2} - \phi)p_{h+1} - \frac{\phi\bar{L}_{h+1}^2}{\gamma^2}p_{h+1}^2\right) \log Q_{h+1}(\beta_h)$$

$$- \left(u + \phi(p - p_h)\right)\left(1 + \frac{p_{h+1}\bar{L}_{h+1}^2}{\gamma^2}\right)\left(\frac{\bar{L}_h^2 p_h}{\bar{L}_{h+1}^2 p_{h+1}} + \frac{\gamma^2 \bar{m}_h}{\bar{L}_{h+1}^2 p_{h+1}}\right) \cdot \log Q_h(\beta_h)$$

$$= w \log \frac{Q_{h+1}(\beta_h)}{Q_h(\beta_h)^{M_{1h}}} + \left(\phi\left(M_{1h} - 1 + \frac{p_h - p}{p_{h+1}}\right) - \frac{u}{p_{h+1}}\right) \log Q_h(\beta_h)^{p_h}$$

$$+ \left(w\frac{\bar{L}^2}{\gamma^2} - \phi\right) \log \frac{Q_{h+1}(\beta_h)^{p_{h+1}}}{Q_h(\beta_h)^{p_h}} - \frac{\phi\bar{L}^2}{\gamma^2} \log \frac{Q_{h+1}(\beta_h)^{p_{h+1}^2}}{Q_h(\beta_h)^{p_h^2}}.$$

*Similarly, for term A, we have*

$$\log_{\psi_H} Q_H(\beta_H)^{\Phi_H} \leq \frac{p_H \bar{L}_H^2 + \gamma^2}{\gamma^2} \log Q_H(\beta_H)^{\Phi_H} = \frac{p_H \bar{L}_H^2 + \gamma^2}{\gamma^2}(u + \phi p - \phi p_H) \log Q_H(\beta_H)^{-1}$$

$$\leq \left(w + (w\frac{\bar{L}^2}{\gamma^2} - \phi)p_H - \frac{\phi \bar{L}^2}{\gamma^2}p_H^2\right) \log Q_H(\beta_H)^{-1}.$$

*Then the following inequality holds true:*

$$T_a \leq \overbrace{w \log \Pi_{h=1}^{H-1} \frac{Q_{h+1}(\beta_h)}{Q_h(\beta_h)^{M_{1h}}} \frac{1}{Q_H(\beta_H)}}^{T_{a1}} + \overbrace{\sum_{h=1}^{H-1} \left(\phi\left(M_{1h} - 1 + \frac{p_h - p}{p_{h+1}}\right) - \frac{u}{p_{h+1}}\right) \log Q_h(\beta_h)^{p_h}}^{T_{a2}}$$

$$+ \overbrace{\left(w\frac{\bar{L}^2}{\gamma^2} - \phi\right) \log \Pi_{h=1}^{H-1} \frac{Q_{h+1}(\beta_h)^{p_{h+1}}}{Q_h(\beta_h)^{p_h}} \frac{1}{Q_H(\beta_H)^{p_H}}}^{T_{a3}} - \overbrace{\frac{\phi \bar{L}^2}{\gamma^2} \log \Pi_{h=1}^{H-1} \frac{Q_{h+1}(\beta_h)^{p_{h+1}^2}}{Q_h(\beta_h)^{p_h^2}} \frac{1}{Q_H(\beta_H)^{p_H^2}}}^{T_{a4}},$$

*where*

$$M_{1h} = m_h + \bar{m}_h = m_h + \frac{1}{1 - \frac{m_h}{2}} = m_h + 1 + \frac{m_h}{2} + O(\frac{m_h^2}{2^2})$$

$$= 1 + \frac{3}{2}m_h + O(\frac{m_h^2}{2^2}) = 1 + \frac{3\gamma^2}{2\bar{L}^2 p_h} + O(\frac{\gamma^4}{4\bar{L}^4 p_h^2}),$$

*and*

$$\phi\left(M_{1h} - 1 + \frac{p_h - p}{p_{h+1}}\right) - \frac{u}{p_{h+1}} = \phi\left(\frac{\gamma^2}{\bar{L}^2 p_h} + O(\frac{\gamma^4}{4\bar{L}^4 p_h^2}) + \frac{p_h - p}{p_{h+1}}\right) - \frac{u}{p_{h+1}}.$$

*Because*

$$T_{a1} = w \log \Pi_{h=1}^{H-1} \frac{Q_{h+1}(\beta_h)}{Q_h(\beta_h)^{M_{1h}}} \frac{1}{Q_H(\beta_H)}$$

$$= w\left(\sum_{h=1}^{H-1} \log \frac{Q_{h+1}(\beta_h)}{Q_h(\beta_h)} + \sum_{h=1}^{H-1} (\frac{3\gamma^2}{2\bar{L}^2 p_h} + O(\frac{\gamma^4}{4\bar{L}^4 p_h^2})) \log Q_h(\beta_h)^{-1} + \log Q_H(\beta_H)^{-1}\right),$$

*and*

$$T_{a2} = \sum_{h=1}^{H-1} \left(\phi\left(M_{1h} - 1 + \frac{p_h - p}{p_{h+1}}\right) - \frac{u}{p_{h+1}}\right) \log Q_h(\beta_h)^{p_h}$$

$$\leq \sum_{h=1}^{H-1} \left(\phi\left(\frac{3\gamma^2}{2\bar{L}^2} + O(\frac{\gamma^4}{4\bar{L}^4 p_h}) + p_h - p\right) - u\right) \log Q_h(\beta_h)$$

$$= \sum_{h=1}^{H-1} \left(\phi\left(p - p_h - \frac{3\gamma^2}{2\bar{L}^2} - O(\frac{\gamma^4}{4\bar{L}^4 p_h})\right) + u\right) \log Q_h(\beta_h)^{-1}$$

$$\leq \sum_{h=1}^{H-1} \left(\phi\left(p - p_h\right) + u\right) \log Q_h(\beta_h)^{-1},$$

*we can derive that*

$$
\begin{aligned}
T_{a1} + T_{a2} \leq & w\bigg( \sum_{h=1}^{H-1} \log \frac{Q_{h+1}(\beta_h)}{Q_h(\beta_h)} + \log Q_H(\beta_H)^{-1} \bigg) + \sum_{h=1}^{H-1} \Big( \phi\big(p - p_h\big) \\
& + u + \frac{3w\gamma^2}{2\bar{L}^2 p_h} + O(\frac{w\gamma^4}{4\bar{L}^4 p_h^2}) \Big) \log Q_h(\beta_h)^{-1} \\
= & (u + \phi p) \sum_{h=1}^{H-1} \log \frac{Q_{h+1}(\beta_h)}{Q_h(\beta_h)} + (u + \phi p) \log Q_H(\beta_H)^{-1} + \\
& \sum_{h=1}^{H-1} \Big( u + \phi\big(p - p_h\big) + \frac{3\gamma^2(u + \phi p)}{2\bar{L}^2 p_h} + O(\frac{\gamma^4(u + \phi p)}{4\bar{L}^4 p_h^2}) \Big) \log Q_h(\beta_h)^{-1}. \quad (6)
\end{aligned}
$$

$$
\begin{aligned}
T_{a3} = & \big(w\frac{\bar{L}^2}{\gamma^2} - \phi\big) \log \Pi_{h=1}^{H-1} \frac{Q_{h+1}(\beta_h)^{p_{h+1}}}{Q_h(\beta_h)^{p_h}} \frac{1}{Q_H(\beta_H)^{p_H}} \\
= & \big(w\frac{\bar{L}^2}{\gamma^2} - \phi\big) \bigg( \sum_{h=1}^{H} p_h \log \frac{Q_{h+1}(\beta_{h+1})}{Q_h(\beta_h)} + \log \frac{\Pi_{h=1}^{H-1} Q_{h+1}(\beta_{h+1})^{p_{h+1} - p_h}}{Q_H(\beta_H)^{p_H}} \bigg) \\
\leq & w\frac{\bar{L}^2}{\gamma^2} \bigg( \sum_{h=1}^{H} p_h \log \frac{Q_{h+1}(\beta_{h+1})}{Q_h(\beta_h)} + \log \frac{\Pi_{h=1}^{H-1} Q_{h+1}(\beta_{h+1})^{p_{h+1} - p_h}}{Q_H(\beta_H)^{p_H}} \bigg). \quad (7)
\end{aligned}
$$

$$
\begin{aligned}
T_{a4} = & \frac{\phi \bar{L}^2}{\gamma^2} \log \Pi_{h=1}^{H-1} \frac{Q_{h+1}(\beta_h)^{p_{h+1}^2}}{Q_h(\beta_h)^{p_h^2}} \frac{1}{Q_H(\beta_H)^{p_H^2}} \\
= & \frac{\phi \bar{L}^2}{\gamma^2} \bigg( \sum_{h=1}^{H} p_h^2 \log \frac{Q_{h+1}(\beta_{h+1})}{Q_h(\beta_h)} + \log \frac{\Pi_{h=1}^{H-1} Q_{h+1}(\beta_{h+1})^{p_{h+1}^2 - p_h^2}}{Q_H(\beta_H)^{p_H^2}} \bigg).
\end{aligned}
$$

*As $T_{a4} < T_{a3}$, $T_{a3} - T_{a4} < T_{a3}$. Then*

$$
\begin{aligned}
T_a < & T_{a1} + T_{a2} + T_{a3} \\
= & (u + \phi p) \sum_{h=1}^{H-1} \log \frac{Q_{h+1}(\beta_h)}{Q_h(\beta_h)} + (u + \phi p) \log Q_H(\beta_H)^{-1} \\
& + \sum_{h=1}^{H-1} \Big( u + \phi\big(p - p_h\big) + \frac{3\gamma^2(u + \phi p)}{2\bar{L}^2 p_h} + O(\frac{\gamma^4(u + \phi p)}{4\bar{L}^4 p_h^2}) \Big) \log Q_h(\beta_h)^{-1} \\
& + w\frac{\bar{L}^2}{\gamma^2} \bigg( \sum_{h=1}^{H} p_h \log \frac{Q_{h+1}(\beta_{h+1})}{Q_h(\beta_h)} + \log \frac{\Pi_{h=1}^{H-1} Q_{h+1}(\beta_{h+1})^{p_{h+1} - p_h}}{Q_H(\beta_H)^{p_H}} \bigg).
\end{aligned}
$$

*Since the first term is smaller than $T_{a3}$ , we have the complexity for the feature recruiting phase as*

$$
\begin{aligned}
T_a < & T_{a1} + T_{a2} + T_{a3} \\
= & O\bigg( \sum_{h=1}^{H} \big( u + \phi\big(p - p_h\big) \big) \log Q_h(\beta_h)^{-1} + \\
& w\frac{\bar{L}^2}{\gamma^2} \Big( \sum_{h=1}^{H} p_h \log \frac{Q_{h+1}(\beta_{h+1})}{Q_h(\beta_h)} + \log \frac{\Pi_{h=1}^{H-1} Q_{h+1}(\beta_{h+1})^{p_{h+1} - p_h}}{Q_H(\beta_H)^{p_H}} \Big) \bigg).
\end{aligned}
$$

*With*

$$
\bar{Q} = \big( \Pi_{h=1}^{H-1} Q_{h+1}(\beta_h)^{p_{h+1} - p_h} \big)^{\frac{1}{p_H}},
$$

$$\mathcal{U} = \log\big(\Pi_{h=1}^{H-1} \frac{Q_{h+1}(\beta_h)}{Q_h(\beta_h)^{\frac{p_h}{p_{h+1}}}} \frac{1}{Q_H(\beta_H)}\big),$$

and

$$\Phi = \log\big(\Pi_{h=1}^{H-1} \frac{Q_{h+1}(\beta_d)^{p_h}}{Q_h(\beta_h)^{p_h}}\big),$$

we get the complexity for feature recruiting as

$$O\bigg( \big(u + p\big(\frac{n\varsigma}{K_1} + \frac{n(1-\varsigma)}{K_1 K_2}\big)\big)\big(\mathcal{U} + \frac{\bar{L}^2}{\gamma^2}\Phi + p_H \frac{\bar{L}^2}{\gamma^2}\log\frac{\bar{Q}}{Q_H(\beta_H)}\big)\bigg).$$

$\square$

## Appendix-C. Convergence of Feature Screening

After Thunder sets DoRecruit to False, usually there are some inactive features remaining in $\mathcal{A}_t$. They can be removed from the active set $\mathcal{A}_t$ with the screening operation. Let $G_d = P(\beta_d) - P(\beta^*)$ represent the primal accuracy for the screening of the $d$-th feature. $\bar{\mathcal{A}}$ is the optimal active feature set of the original LASSO problem that $\{x_i : |x_i^\top \theta^*| = 1\}$. Let $G_{p_H} = Q_{p_H}$, the complexity for the feature screening phase is given by Lemma 5.

**Lemma 5** *Let $Z_D$ be the total number of features removed from the active set after DoRecruit is set to False. The upper bound of the complexity for feature screening phase is*

$$u\log\frac{G_{p_H}}{\varepsilon} + \frac{u\bar{L}^2}{\gamma^2}\Big((p_H - |\bar{\mathcal{A}}|)\log\frac{G_{p_H}}{G_{Z_D}} + |\bar{\mathcal{A}}|\log\frac{G_{p_H}}{\varepsilon}\Big) + \frac{n}{K_1}\Big(p_H\log\frac{G_{p_H}}{G_{Z_D}} + \frac{\bar{L}^2}{\gamma^2}p_H^2\log\frac{G_{p_H}}{G_{Z_D}}\Big).$$

**Proof**: *To prove Lemma 5, we can use the similar strategies as for Lemma 4. We need to add up the operations needed to reach the screening accuracy threshold for all the inactive features. Let $T_b$ denotes the time consumed by both inactive feature screening and accuracy pursuing phases. $p_d$ is the size of feature set after $d$ features have been removed with the screening procedure. We have*

$$T_b = \sum_{d=1}^{Z_D} \frac{\log_{\psi_{d-1}} \frac{G_d}{G_{d-1}}}{K_1}(K_1 u + n p_{d-1}) + u\log_{\psi_{Z_D}} \frac{\varepsilon}{G_{Z_D}}$$

$$= \overbrace{u\sum_{d=1}^{Z_D} \log_{\psi_{d-1}} \frac{G_d}{G_{d-1}} + u\log_{\psi_{Z_D}} \frac{\varepsilon}{G_{Z_D}}}^{T_{b1}} + \overbrace{\frac{n}{K_1}\sum_{d=1}^{Z_D} p_{d-1}\log_{\psi_{d-1}} \frac{G_d}{G_{d-1}}}^{T_{b2}}.$$

*The first two terms can be written as*

$$T_{b1} = u\sum_{d=1}^{Z_D} \log_{\psi_{d-1}} \frac{G_d}{G_{d-1}} + u\log_{\psi_{Z_D}} \frac{\varepsilon}{G_{Z_D}}$$

$$\leq u\log\frac{G_{p_H}}{\varepsilon} + \frac{u\bar{L}^2}{\gamma^2}\Big((p_H - |\bar{\mathcal{A}}|)\log\frac{G_{p_H}}{\bar{G}} + |\bar{\mathcal{A}}|\log\frac{G_{p_H}}{\varepsilon}\Big).$$

*Here $\bar{G} = \big(\Pi_{d=1}^{Z_D} G_d\big)^{\frac{1}{p_H - |\bar{\mathcal{A}}|}}$.*

$$T_{b2} = \frac{n}{K_1}\sum_{d=1}^{Z_D} p_{d-1}\log_{\psi_{d-1}} \frac{G_d}{G_{d-1}}$$

$$\leq \frac{n}{K_1}\sum_{d=1}^{Z_D}(p_{d-1} + \frac{p_{d-1}^2 \bar{L}^2}{\gamma^2})\log\frac{G_{d-1}}{G_d}$$

$$= \frac{n}{K_1}\Big(\log\frac{G_{p_H}^{p_H}}{\bar{G}^{p_H - |\bar{\mathcal{A}}|}G_{Z_D}^{|\bar{\mathcal{A}}|}} + \frac{\bar{L}^2}{\gamma^2}\log\frac{G_{p_H}^{p_H^2}}{\tilde{G}^{p_H^2 - |\bar{\mathcal{A}}|^2}G_{Z_D}^{|\bar{\mathcal{A}}|^2}}\Big),$$

*where $\tilde{G} = \left(\Pi_{d=1}^{Z_D} G_d^{p_{d-1}^2 - p_d^2}\right)^{\frac{1}{p_H^2 - |\bar{\mathcal{A}}|^2}}$.*

   *Since*

$$\bar{G} \geq \left(\Pi_{d=1}^{Z_D} G_{Z_D}\right)^{\frac{1}{p_H - |\bar{\mathcal{A}}|}} = G_{Z_D},$$

   *and*

$$\tilde{G} \geq \left(\Pi_{d=1}^{Z_D} G_{Z_D}^{p_{d-1}^2 - p_d^2}\right)^{\frac{1}{p_H^2 - |\bar{\mathcal{A}}|^2}} = G_{Z_D},$$

   *we get*

$$T_{b1} \leq u \log \frac{G_{p_H}}{\varepsilon} + \frac{u\bar{L}^2}{\gamma^2}\left((p_H - |\bar{\mathcal{A}}|) \log \frac{G_{p_H}}{G_{Z_D}} + |\bar{\mathcal{A}}| \log \frac{G_{p_H}}{\varepsilon}\right), \tag{8}$$

   *and*

$$T_{b2} \leq \frac{n}{K_1}\left(p_H \log \frac{G_{p_H}}{G_{Z_D}} + \frac{\bar{L}^2}{\gamma^2} p_H^2 \log \frac{G_{p_H}}{G_{Z_D}}\right). \tag{9}$$

Thus the upper bound of the complexity for feature screening stage is as stated in the lemma. $\square$

## Appendix-D. Proof of Theorem 1

Based on the analysis in Lemma 4 and 5, the complexity of the proposed method is given by the following theorem.

**Theorem 1** *With $O(u)$ as the complexity for one iteration of coordinate minimization of the LASSO problem with a $\gamma$-convex loss function, the time complexity for the proposed algorithm is $O\left(u\frac{\bar{L}^2}{\gamma^2}\left(\eta\bar{p}\log\frac{\bar{Q}}{\varepsilon_D} + \eta\bar{p}H + |\bar{\mathcal{A}}|\log\frac{\varepsilon_D}{\varepsilon}\right)\right)$. Here $H$ is the total number of features involved in recruiting operations, $\bar{p}$ is the maximum size of the active set during the algorithm iterations, $\bar{Q}$ is the geometric mean of the sub-problem primal objective function precision values corresponding to each recruiting operation, and $\varepsilon_D$ is the primal objective function precision for the last feature screening operation. $\eta = 1 + \frac{np\varsigma}{uK_1} + \frac{np(1-\varsigma)+p\log p}{uK_1K_2}$, and $\varsigma$ is the feature partition ratio for $\mathcal{R}_1$ and $\mathcal{R}_2$.*

**Proof**: *With $G_{p_H} = Q_H(\beta_H)$, $\phi = \frac{n\varsigma}{K_1} + \frac{n(1-\varsigma)+\log p}{K_1K_2}$, based on Lemma 4 and Lemma 5, the time complexity for the proposed algorithm can be written as*

$$T < T_a + T_b = T_{a1} + T_{a2} + T_{a3} + T_{b1} + T_{b2}.$$

*Based on the proofs for Lemma 4 and Lemma 5, according to (7), (8) and (9)*

$T_{a3} + T_b = T_{a3} + T_{b1} + T_{b2}$

$$= (u + p\phi)\frac{\bar{L}^2}{\gamma^2}\left(\sum_{h=1}^{H} p_h \log \frac{Q_{h+1}(\beta_{h+1})}{Q_h(\beta_h)} + \log \frac{\Pi_{h=1}^{H-1} Q_{h+1}(\beta_{h+1})^{p_{h+1}-p_h}}{Q_H(\beta_H)^{p_H}}\right) + u \log \frac{G_{p_H}}{\varepsilon}$$

$$+ \frac{u\bar{L}^2}{\gamma^2}\left((p_H - |\bar{\mathcal{A}}|) \log \frac{G_{p_H}}{G_{Z_D}} + |\bar{\mathcal{A}}| \log \frac{G_{p_H}}{\varepsilon}\right) + \frac{n}{K_1}\left(p_H \log \frac{G_{p_H}}{G_{Z_D}} + \frac{\bar{L}^2}{\gamma^2} p_H^2 \log \frac{G_{p_H}}{G_{Z_D}}\right)$$

$$\leq (u + p\phi)\left(\frac{\bar{L}^2}{\gamma^2}\Phi + p_H\frac{\bar{L}^2}{\gamma^2}\log\frac{\bar{Q}}{Q_H(\beta_H)}\right) + u\log\frac{G_{p_H}}{\varepsilon} + \frac{u\bar{L}^2}{\gamma^2}\left((p_H - |\bar{\mathcal{A}}|)\log\frac{G_{p_H}}{G_{Z_D}} + |\bar{\mathcal{A}}|\log\frac{G_{p_H}}{\varepsilon}\right) +$$

$$\frac{n}{K_1}\left(p_H\log\frac{G_{p_H}}{G_{Z_D}} + \frac{\bar{L}^2}{\gamma^2}p_H^2\log\frac{G_{p_H}}{G_{Z_D}}\right)$$

$$= (u + p\phi)\left(\frac{\bar{L}^2}{\gamma^2}\Phi + p_H\frac{\bar{L}^2}{\gamma^2}\log\frac{\bar{Q}}{G_{p_H}}\right) + u\log\frac{G_{p_H}}{\varepsilon} + \frac{u\bar{L}^2}{\gamma^2}\left(p_H\log\frac{G_{p_H}}{G_{Z_D}} + |\bar{\mathcal{A}}|\log\frac{G_{Z_D}}{\varepsilon}\right)$$

$$+ \frac{np_H}{K_1}\left(\log\frac{G_{p_H}}{G_{Z_D}} + \frac{\bar{L}^2}{\gamma^2}p_H\log\frac{G_{p_H}}{G_{Z_D}}\right)$$

$$\leq p_H\left(u + \frac{n\varsigma}{C} + \frac{n(1-\varsigma)+\log p}{CK_2}\right)\frac{\bar{L}^2}{\gamma^2}\log\frac{\bar{Q}}{G_{Z_D}} + u\frac{\bar{L}^2}{\gamma^2}|\bar{\mathcal{A}}|\log\frac{G_{Z_D}}{\varepsilon} + (u + \frac{n}{C})\frac{\bar{L}^2}{\gamma^2}\Phi +$$

$$u\log\frac{G_{p_H}}{\varepsilon} + \frac{n}{C}\log\frac{G_{p_H}}{G_{Z_D}}.$$

*Here* $C = K_1/p$. *Let* $\eta = 1 + \frac{n\varsigma}{uC} + \frac{n(1-\varsigma)+\log p}{uCK_2}$, $\bar{p} = \max_{h:1\leq h\leq H} p_h$, *and* $c_1 = \max_{h:1\leq h\leq H-1} \log \frac{Q_{h+1}(\beta_h)}{Q_h(\beta_h)}$, *then we have*

$$(u + \frac{n}{C})\frac{\bar{L}^2}{\gamma^2}\Phi$$

$$\leq u(\eta - \frac{n(1-\varsigma)+\log p}{uCK_2})\frac{\bar{L}^2}{\gamma^2}\log\left(\Pi_{h=1}^{H-1}\frac{Q_{h+1}(\beta_h)^{p_h}}{Q_h(\beta_h)^{p_h}}\right)$$

$$\leq u\eta\frac{\bar{L}^2}{\gamma^2}\bar{p}Hc_1.$$

*According to* (6),

$$T_{a1} + T_{a2} \leq c_2 \sum_{h=1}^{H}\left(u + \phi p\right)\log Q_h(\beta_h)^{-1}$$

$$\leq c_2 c_3 u\eta H,$$

*where* $c_3 = \max_{h:1\leq h\leq H} \log Q_h(\beta_h)^{-1}$ *and* $0 < c_2 < 2$.

*Thus*

$$T < u\eta p_H\frac{\bar{L}^2}{\gamma^2}\log\frac{\bar{Q}}{G_{Z_D}} + u\frac{\bar{L}^2}{\gamma^2}|\bar{\mathcal{A}}|\log\frac{G_{Z_D}}{\varepsilon} + u\eta\frac{\bar{L}^2}{\gamma^2}\bar{p}Hc_1 + c_2 c_3 u\eta H + u\log\frac{G_{pH}}{\varepsilon} + \frac{n}{C}\log\frac{G_{pH}}{G_{Z_D}}.$$

*Let* $\epsilon_D = G_{Z_D}$, *the time complexity for the proposed algorithm can be simplified as* $O\left(u\frac{\bar{L}^2}{\gamma^2}\left(\eta\bar{p}\log\frac{\bar{Q}}{\varepsilon_D} + \eta\bar{p}H + |\bar{\mathcal{A}}|\log\frac{\varepsilon_D}{\varepsilon}\right)\right)$. *Here* $\eta = 1 + \frac{np\varsigma}{uK_1} + \frac{np(1-\varsigma)+p\log p}{uK_1K_2}$. $\quad\square$

According to the proof of Theorem 1, the algorithm complexity is given by

$$O\left(u\frac{\bar{L}^2}{\gamma^2}\left(\eta\bar{p}\log\frac{\bar{Q}}{\varepsilon_D} + c_1\eta\bar{p}H + |\bar{\mathcal{A}}|\log\frac{\varepsilon_D}{\varepsilon}\right)\right).$$

Here $c_1 = \frac{1}{H-1}\log\frac{\Pi_{i=1}^{H-1}Q_{h+1}(\beta_h)}{\Pi_{i=1}^{H-1}Q_h(\beta_h)} = \frac{1}{H-1}\log\frac{\Pi_{i=1}^{H-1}Q_{h+1}(\beta_h)}{\Pi_{i=1}^{H-1}\left(Q_h(\beta_{h-1})-K_1 d_h\right)}$, and $d_h$ is the average step size of the primal sub-problem. With $\eta = 1 + \frac{np\varsigma}{uK_1} + \frac{np(1-\varsigma)+p\log p}{uK_1K_2}$, after some calculation, we obtain the optimal approximation of $K_1$ given by $a\sqrt{np/u}$, where $a$ is a constant value. In the algorithm, we can set $K_1$ proportional to $\sqrt{np/u}$. Experimentally, the performance of Thunder is not sensitive to the value of $K_2$.