[Reviews · NeurIPS 2020]

Review 1

Summary and Contributions: This paper introduces a new fast solver for solving large scale l-1 constrained minimization problems (aka Lasso) called Thunder. The authors propose a dual stage minimization procedure which consists of screening and recruiting of relevant support (feature selection) followed by shrinkage. They provide theoretical analysis of this algorithm and also demonstrate its efficiency as a solver on both simulated as well as real datasets, w.r.t. Celer and BLITZ. The feature set F is decomposed into an active set A and remaining set R, and Thunder solves sub-problems to actively refine A until a specifically designed stopping condition is satisfied.

Strengths: Pre-existing approaches for fast sparse solvers include feature screening for Lasso [10], which develop rules based on dual variable \lambda to refine the feature set of the sparse vectors pre-shrinkage thereby reducing computation time. Other approaches rely on correctly adjusting dual variable \lambda (homotopy based) and solving sub-problems (working set method) to achieve better computational efficiency. This paper effectively combines the merits of all of these approaches and packages it as a multi-stage sparse solver.

Weaknesses: The paper can benefit with a better exposition on safe screening methods [10] (lines 17-23). Line 26: "unsafe assumptions" vs "safe" screening rules - either explain briefly or defer explanation to a later point.

Correctness: The claims appear to be correct.

Clarity: Yes.

Relation to Prior Work: Yes.

Reproducibility: Yes

Additional Feedback:


Review 2

Summary and Contributions: The paper introduces a solver for lasso-type problems with a new screening strategy. The screening strategy is based on operational properties of the active set, with feature recruiting and removing components. The recruiting proposed is equipped with sampling approximation and a two-layer splitting selection. Numerical comparison on synthetic and application data sets demonstrates the advantage of the method with respect to timing. The major contribution of the paper is the screening strategy with its two-layer recruiting set. The design of this improved strategy is based on insightful comparison with a few currently popular screening methods. Rebuttal checked. Agree with other reviews and will keep the score.

Strengths: The design of the strategy is clear and well-motivated by the limitations of working set methods and dynamic screening. The empirical performance is promising.

Weaknesses: The details of the algorithm may need to be better presented and discussed. The effects of the a few proposed components are not clearly demonstrated. For example, what is the impact of the sampling recruiting strategy in 2.2 on both accuracy, convergence properties and speed? Does it break the safe screen property? Similarly, what is the effect of the size ratio of 2.3 on the algorithm, theoretically and empirically? Since these are the major innovative components, it would be to study it in details. For example, in the experiments, fix/remove the 2.3 and only do 2.2, and evaluate the difference. The current experiments are not informative. The broad impact part is not very informative -- it is essentially a summary of introduction, instead of board impact statement.

Correctness: Did not check the supplement proof. The main text looks correct.

Clarity: The writting is reasonable. Some improvement can be achieved by using more consistent notations with clear definitions. The algorithm description should also be improved.

Relation to Prior Work: Yes

Reproducibility: Yes

Additional Feedback:


Review 3

Summary and Contributions: This paper proposes a new solver, a.k.a, the Thunder solver, for large-scale l1 regularized optimization. In addition, this paper also provides theoretical justification and experiments for the Thunder algorithm in the proposed solver.

Strengths: This work proposes a novel solver for a general l1 regularized convex optimization problem with the L-Lipschitz gradient. Experimentally, the Thunder solver is faster than the recent Celer solver and BlitzL1 solver. Theoretically, this paper provides nontrivial time complexity analysis for the proposed algorithm.

Weaknesses: 1) This work should have mentioned more related works. I.e., when introducing the l1 regularization for sparse learning, compressed sensing, some classic works such as https://arxiv.org/pdf/math/0502327.pdf, https://authors.library.caltech.edu/10092/1/CANieeespm08.pdf, https://www.cns.nyu.edu/~tony/vns/readings/olshausen-field-1996.pdf, https://www.sciencedirect.com/science/article/pii/S0042698997001697, https://statweb.stanford.edu/~tibs/lasso/lasso.pdf, should be cited. 2) In the experiment section, it would appeal to a broader audience if the paper could explain the relationship between Thunder and CVX solver and provide some experimental comparison, since people in other fields may be unfamiliar with the Celer solver and BlitzL1 solver. 3) Although the experiments show some practical benefits, I can not find any principled way to choose the hyperparameters such as K_1, K_2. Are their choices problem specific? Are there similar problem specific parameters for Celer and Blitz optimized for fair comparison? The improvements of Thunder seem to come from more careful engineering of the work sets. Any more fundamental reasons that are attributed to the improvements?

Correctness: The theoretical results seem sound and correct and the experiments seem to be reasonable.

Clarity: The paper is well-organized and presented.

Relation to Prior Work: Yes, the novelty is clearly discussed. If the comparison is fair and not case by case, then the proposed algorithm could improve state of the art for L1 solvers. But at time of the review, this is difficult to assess.

Reproducibility: Yes

Additional Feedback: 1) Bad wording: "seminal", the second word of line 25 2) The labels in pictures are small (Figure 4, 5, 7, 8)


Review 4

Summary and Contributions: This paper proposed a novel lasso screening methods. By learning an active set which is much smaller the whole feature set, the efficiency is greatly improved. The most expensive part is the feature recruiting, the author uses a bi-level selection strategy to further improve the efficiency. Besides efficiency, the author also theoretically shows the safeness of the algorithms. In complexity analysis, the author pointed out the complexity can be largely reduced when the ground truth sparsity ratio or sparse penalty is large, which is the theoretical highlights. The authors have done several experiments to support their claims.

Strengths: The efficiency is largely improved compared to the previous study. The theoretical results and empirical results are consistent. Algorithms are not complex, thus easy to be applied.

Weaknesses: The complexity is largely reduced by feature shrinking. How does the ratio between R_1 and R_2 and the shrinking frequency K_2 affect the final solution? There is no sensitivity analysis on this part. Does the proposed approach require feature independence? Since the author mentioned that the active set will be the same size as the optimal set, is it possible that this will hold when features are correlated? This is a minor suggestion. Since the real data may violate the assumption, except for the running time, it is better to add the prediction results to show whether the right feature is selected.

Correctness: Yes, both empirical results and theoretical results are sound.

Clarity: yes, the logic is clear.

Relation to Prior Work: yes, the author uses a subset selection to avoid large-scale computation.

Reproducibility: Yes

Additional Feedback: Generally, the paper is impressive in terms of both theoretical and empirical aspects. My main concerns are as follows: 1). The assumptions: in the empirical study, the author highlights that compared to celer, the active set is more close to the optimal active set. Is it because that celer includes the correlated features? I wish to know whether the proposed method really found those active features but the author does not provide the results for this. Especially for real data, when the features are highly correlated, can the method improved the post-stage lasso prediction results? Also, the Lipstchiz and convexity ratio might lead to numerical issue in real dataset, which could effect the prediction results. If the author includes prediction in results, I think the paper will be more solid. 2). Splitting the R_1 and R_2. Since the author chooses 1/3 as the ratio in the experiment, I am wondering whether we can be more aggressive on this. The sensitivity analysis of the ratio and shrinking frequency is missing in the paper.

[Author Response · NeurIPS 2020]

We sincerely thank the reviewers for their time and constructive comments. We try to focus on one point raised by the
reviewers in each paragraph as follows.

The proposed algorithm in this paper is to improve the solution efficiency of the sparse learning problems given by
equation (1) in the main file. As discussed at the beginning of the supplemental file, Thunder outperforms existing
solvers is mainly because of the passive feature recruiting strategies, sampling method for feature recruiting, and the
safe stop condition regarding feature recruiting employed by the algorithm. These strategies can ensure Thunder has
a smaller active set during algorithm updating, effective feature screening, efficient feature recruiting, and algorithm
safety guarantee. According to our complexity study in Theorem 1 and Section 3.2, maintaining a small active set is
crucial to active/working set type of algorithms. The efficiency of Thunder is based on strong theoretical support rather
than engineering tricks.

The prediction and feature selection accuracy relies on the selection of $\lambda$. This $\lambda$ selection problem has been studied by
the statistics community, and it is beyond the scope of this paper. In this paper, we focus on optimization methodologies
that can further scale up the solutions of sparse learning given one particular $\lambda$. As the problem is convex, duality gap is
usually used to measure the precision of the solutions regarding a particular $\lambda$ value.

The correlation between features may affect the efficiency of Thunder, but it does not impact the algorithm's safety.
Here safety means the algorithm final step active set does not miss any features in the optimal active set $\bar{\mathcal{A}}$ of the
problem. According to the derivation in Section 2.1, the stop condition regarding feature recruiting given in Lemma 1
ensures that the final active set is a super set that contains the optimal active set. If the condition in Lemma 1 is not
met, Algorithm 2 will not stop feature recruiting. Each operation and updating of Algorithm 2 will decrease (or not
change) the duality gap of the original problem, and the problem is convex. The duality gap will become smaller and
smaller and then the algorithm can distinguish all active features according to Lemma 1. The safety of the algorithm is
guaranteed by the safety of the operation at each step. As shown in the experiments, Thunder can outperform existing
solvers on all three large real-world data sets, i.e., Finance, KDD2010, and URL. The results on these real-world data
sets prove the advantages and effectiveness of Thunder under different data correlation scores.

According to the proof of Theorem 1, the algorithm complexity is given by $O\left( u\frac{\bar{L}^2}{\gamma^2}\left(\eta\bar{p}\log\frac{\bar{Q}}{\varepsilon_D}+c_1\eta\bar{p}H+|\bar{\mathcal{A}}|\log\frac{\varepsilon_D}{\varepsilon}\right)\right)$.

Here $c_1 = \frac{1}{H-1}\log\frac{\Pi_{i=1}^{H-1}Q_{h+1}(\beta_h)}{\Pi_{i=1}^{H-1}Q_h(\beta_h)} = \frac{1}{H-1}\log\frac{\Pi_{i=1}^{H-1}Q_{h+1}(\beta_h)}{\Pi_{i=1}^{H-1}\left(Q_h(\beta_{h-1})-K_1 d_h\right)}$, and $d_h$ is the average step size of the primal

sub-problem. With $\eta = 1 + \frac{np\varsigma}{uK_1} + \frac{np(1-\varsigma)+p\log p}{uK_1K_2}$, after derivation we can get the optimal approximation of $K_1$ given
by $a\sqrt{np/u}$, and $a$ is a constant value. In the algorithm, we can set $K_1$ proportional to $\sqrt{np/u}$. Experimentally, the
performance of Thunder is not sensitive to the value of $K_2$. We agree with the reviewers that we will include detailed
theoretical analysis as well as the experimental study regarding the selection of $K_1$ and $K_2$ in the next version.

Similarly, in our experiments, the feature partition ratio $\varsigma$ does not affect Thunder's performance very significantly. As
long as the size of $\mathcal{R}_t^1$ is more than around 1.5 times of $\mathcal{A}_t$, the performance of Thunder does not change a lot regarding
$\varsigma$. Thunder is not very sensitive to either $\varsigma$ or $K_2$ is because that the operations on $\mathcal{A}_t$ and $\mathcal{R}_t^1$, and the inner loop
updating takes the main part of the algorithm. Another reason is that the sampling strategy utilized by Thunder can
significantly reduce the feature recruiting and condition checking complexity resulted from the features outside of $\mathcal{A}_t$.
The current algorithm complexity analysis in the supplemental file ignores the sampling steps. We will improve the
complexity analysis along with the detailed sensitivity study regarding $\mathcal{R}_t^1$ and $\mathcal{R}_t^2$ ratio in the next version.

To recruit an active feature $x_i \in \mathcal{R}_t^1$, we need to evaluate its activity with $|x_i^\top\theta^*|$. However, here $\theta^*$ is unknown optimal
dual variable, we have to use the current $\theta_t$ in hand to approximate the feature's activity. As mentioned above, we
employ passive feature recruiting strategies, and it means that we only perform the recruiting operation when we are
pretty sure about the features' activity. Give a feature $x_i \in \mathcal{R}_t^1$, if its activity ($|x_i^\top\theta_t|$) lower bound is larger than
the upper bounds of most features in $\mathcal{R}_t^1$, we can say that we are confident about its activity, and then move it to the
active set $\mathcal{A}_t$. The purpose of the proposed sampling strategy is to reduce the cost induced by the condition checking
step in the feature recruiting operation. Instead of comparing the lower bound of $|x_i^\top\theta_t|$ with most feature's upper
bound, we do the comparison with a small subset of it. The sampling strategy does not reduce or break the algorithm's
accuracy and safety, and it is because that the algorithm's safety is guaranteed by the safe stop condition regarding
feature recruiting. We will take the reviewers' suggestions and show more results on the effectiveness of sampling.

We thank the reviewers again for their insightful comments on writing. We will improve the figures, descriptions of the
algorithm, term definition, notations, and writing based on their suggestions. We will include the papers listed by the
reviewers in the reference.

[Meta-Review · NeurIPS 2020]

The reviewers were happy with the paper. It would be great if you include the promised changes + discussion from the rebuttal in the final version of the paper.